# Temporal Change Sensitive Representation for Reinforcement Learing

## Abstract

Image-based deep reinforcement learning has made a great improvement recently by combining state-of-the-art reinforcement learning algorithms with self-supervised representation learning algorithms. However, these self-supervised representation learning algorithms are designed to preserve global visual information, which may miss changes in visual information that are important for performing the task, like in Figure 1. To resolve this problem, self-supervised representation learning specifically designed for better preserving task relevant information is necessary. Following this idea, we introduce Temporal Change Sensitive Representation (TCSR), which is designed for reinforcement learning algorithms that have a latent dynamic model. TCSR enforces the latent state representation of the reinforcement agent to put more emphasis on the part of observation that could potentially change in the future. Our method achieves SoTA performance in Atari100K benchmark.

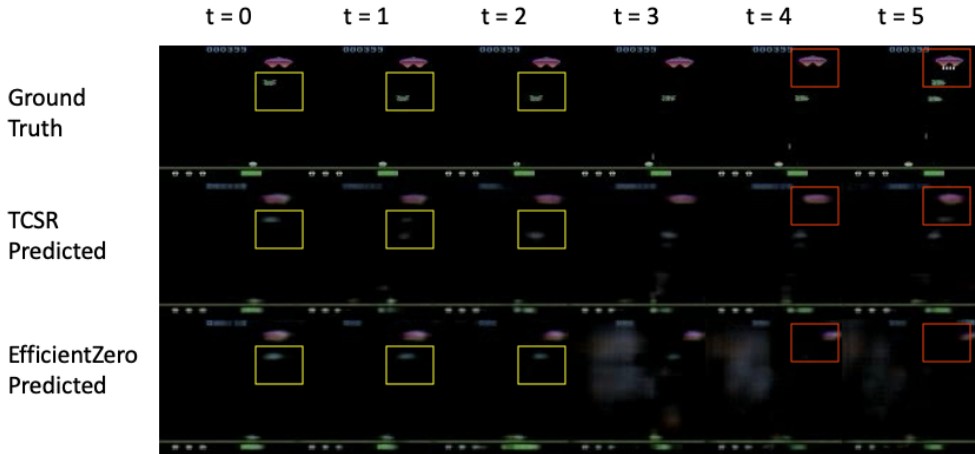

Figure 1: The ground truth observation compared with image reconstructed from latent state representation predicted by TCSR and EfficientZero. TCSR can not only predict the movement of enemies in the short term (Marked in the yellow box) but also predict exactly when and where the UFO will release a new enemy till the end of the planning horizon (Marked in the red box). However, EfficientZero fails to predict both of these changes. This shows that TCSR is more sensitive to the changes in the latent state representation. These change includes but not limited to position, appearance and disappearance of task related objects as shown in this figure.

## 1 Introduction

Deep reinforcement learning has achieved much success in solving image based tasks over the last several years. A critical step to solving image based tasks is learning a good representation of the

image input. One of the biggest challenges for learning a good representation for reinforcement learning is that the reward is sparse (Shelhamer et al., 2016), which cannot generate enough training signal to train the representation network. To resolve this problem, self-supervised representation learning loss is often added to facilitate training.

There are many different approaches to image based reinforcement learning. Most of them try to combine state-of-the-art model based or model free backbones like SAC (Haarnoja et al., 2018), Rainbow (Hessel et al., 2018) and MuZero (Schrittwieser et al., 2020) with self-supervised representation learning algorithms to boost the training of representation. Among these methods, SPR (Schwarzer et al., 2020) and EfficientZero (Ye et al., 2021) are state-of-the-art model-free and model based methods in the Atari 100K benchmark. They achieved the best score in 21 out of 26 Atari 100K games combined. They train a dynamic model to predict the future latent states from an initial latent state calculated by the image encoder. Both the image encoder and the dynamic model are trained using the SimSiam(Chen & He, 2020) loss between the predicted latent state and the latent state calculated directly from the future observations.

However, most representation learning algorithms used in reinforcement learning do not emphasize the change of visual information, while creatures, including humans, are innately sensitive to the change of visual information. A very important part of the neural system is the middle temporal visual area (MT) (Von Bonin & Bailey, 1947). Visual information is integrated and differentiated in MT to capture the movement of objects contained in visual information (Allman et al., 1985). The ability to capture the changes in visual information helps creatures catch prey or escape an enemy Maturana et al. (1960); Suzuki et al. (2019). To help reinforcement learning agents acquire such ability, we propose Temporal Change Sensitive Representation (TCSR), a self-supervised auxiliary loss specifically designed for reinforcement learning methods that have a latent dynamic model. TCSR enforces the difference between two consecutive unrolled latent states to be the same as the difference between two target latent states generated from two consecutive observations with the same augmentation.

TCSR uses EfficientZero (Ye et al., 2021) as the backbone and inherit most of the hyper-parameter. On the Atari 100k benchmark, TCSR surpasses EfficientZero in 19 out of 26 games (as shown in Figure 2) and achieves a new state-of-the-art performance.

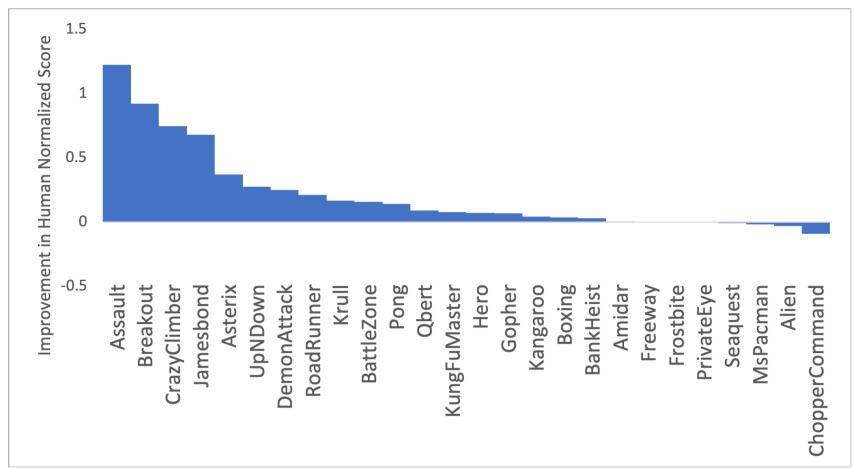

Figure 2: The improvement of human normalized score by adding TCSR as an extra self-supervised representation learning auxiliary loss on EfficientZero backbone. TCSR surpasses EfficientZero in 19 out of 26 games in the Atari 100k benchmark

## 2 RELATED WORK

### 2.1 REPRESENTATION LEARNING IN REINFORCEMENT LEARNING

Almost every image based reinforcement learning algorithms learn a lower dimensional latent state representation from images. Self-supervised auxiliary tasks are often used to facilitate the training of the representation network. Some early works (Lange & Riedmiller, 2010) (Yarats et al., 2019) use widely accepted image reconstruction losses as the auxiliary loss. There was a trend (Srinivas et al., 2020; He et al., 2019a; Banino et al., 2021) of using contrastive losses as auxiliary loss led by CPC (Oord et al., 2018). Recently, similarity based losses (Grill et al., 2020; Chen & He, 2020) are more popular than contrastive losses since they do not need a large number of negative pairs. Temporal consistent/predictive loss is another auxiliary loss that is often used to encourage the representations learned by the agent to contain predictive information (Schwarzer et al., 2020; Guo et al., 2020; Nguyen et al., 2021). Latent state representations learned with temporal consistent/predictive loss could also be unrolled by the dynamic/transitions network to simulate experience (Hafner et al., 2019a; 2020) or perform planning (Hafner et al., 2019b). As the backbone of our algorithm, EfficientZero (Ye et al., 2021) uses the unrolled latent state representation for both training and planning.

### 2.2 MODEL BASED REINFORCEMENT LEARNING

Model based reinforcement learning algorithms usually have access to or learn a world model. Given the current state and a action, the model can predict the next state and the next reward. The model can be used to generate simulated experience and/or perform planning (Sutton & Barto, 2018). Learning a model to generate simulated experience and perform planning with high dimensional inputs like images(Schrittwieser et al., 2020; Ye et al., 2021) is usually more challenging than with low dimensional states (Abbeel et al., 2006; Deisenroth & Rasmussen, 2011). Some reinforcement learning algorithms learn a world model only for training the representation network (Schwarzer et al., 2020; Kaiser et al., 2019; Guo et al., 2020). Dreamer (Hafner et al., 2019a; 2020) only use the learned model to generate simulated experience for training. PlaNet (Hafner et al., 2019b) only use the learned model for planning. Our work aims to help model based reinforcement learning algorithms with high dimensional inputs to train a better model and representation network.

### 2.3 VIDEO PREDICTION

Video prediction has been a classic topic in the field of machine learning (Oprea et al., 2020). Action conditioned video prediction of Atari games could be dated back to 2015 (Oh et al., 2015; Chiappa et al., 2017). They are the foundation work of learning a world model of Atari games for reinforcement learning. Some video prediction algorithms focus on the temporal changes of the inputs (Michalski et al., 2014; Finn et al., 2016), which is similar to our work. But the temporal changes they focus on are at the pixel level while we focus on the latent state level, and our algorithm is designed for reinforcement learning.

## 3 BACKGROUND

### 3.1 MCTS OF MUZERO

MuZero (Schrittwieser et al., 2020) is a Monte-Carlo Tree Search (MCTS) based Reinforcement Learning method. MuZero operates the MCTS with a representation function, a dynamic function and a prediction function. The representation function $h$ encodes an observation $o_t$ into latent state representation $s_{t,0} = h(o_t)$. The dynamic function predicts next latent state representation and reward $s_{t,k+1}, r_{t,k+1} = g(s_{t,k}, a_{t+k})$ given current latent state representation $s_{t,k}$ and action $a_{t+k}$. Given a latent state $s_{t,k}$, the prediction function predicts the policy and value $p_{t,k}, v_{t,k} = f(s_{t,k})$. The policy $p_{t,k}$ is used to expand and navigate through the tree. The value $v_{t,k}$ is used to estimate the values of each node of the tree.

When collecting data, MuZero performs MCTS at each unroll step $t$. The action $a_t$ will be chosen through UCB based on the result of MCTS. Then the action is passed to the environment. The

resulting next observation $o_{t+1}$ and reward $u_{t+1}$ are stored in the replay buffer. The replay buffer also stores the expected return $z_t$ at the root estimated with MCTS and the action distribution $\pi_t$ at the root.

During training, a batch of samples is chosen from the replay buffer. A sample consists of initial observation $o_t$, action sequence $a_t, ...a_{t+K-1}$, ground truth reward sequence $u_{t+1}, ...u_{t+K}$, bootstrapped value target sequence $z_t, ...z_{t+K}$ and action distribution sequence $\pi_t, ...\pi_{t+K}$, where K is the maximum unroll length. Initial state is generated from the stacked observation $s_{t,0} = h(o_{t-n}, ...o_t)$. Unrolled latent state representation and predicted reward are generated recursively with the dynamic network $s_{t,k+1}, r_{t,k+1} = g(s_{t,k}, a_{t+k})$. Predicted policies and values are generated with prediction function $p_{t,k}, v_{t,k} = f(s_{t,k})$ for each unrolled latent state representation. At last, the MuZero loss is calculated as follows:

$$\mathcal{L}_{\text{MuZero}}(t) = \sum_{k \in \{0, ...K\}} \mathcal{L}_{\text{reward}}(u_{t+k}, r_{t,k}) + \mathcal{L}_{\text{value}}(z_{t+k}, v_{t,k}) + \mathcal{L}_{\text{policy}}(\pi_{t+k}, p_{t,k})$$

Where $\mathcal{L}_{\text{reward}}$, $\mathcal{L}_{\text{value}}$, and $\mathcal{L}_{\text{policy}}$ are Cross-Entropy loss. It is critical to notice that MuZero does not assume the predicted latent states correspond to actual states. The predicted latent states generalize across future states that have similar subsequence values, rewards and action distributions (Schrittwieser et al., 2020).

### 3.2 TEMPORAL CONSISTENT LOSS OF EFFICIENTZERO

EfficientZero (Ye et al., 2021) is an efficient sample variant of MuZero that addresses three problems: no supervision on dynamic and representation function, state aliasing and off-policy issue. To address these problems, EfficientZero made three modifications to MuZero. The first modification is adding a self-supervised consistent loss. The second modification is adding an end-to-end prediction of the value prefix by predicting the value prefix with multiple previous unrolled states instead of just the current state. The third modification is model-based off-policy correction, which is done by performing another MCTS tree search at each leaf node to obtain more accurate state value.

Among these three modifications, Self-Supervised Consistent Loss made the most contribution to the final result. Similar to SPR, MuZero's self-supervised loss uses SimSiam self-supervised framework and trains the dynamic function and representation function at the same time. During training, a sequence of observation $o_{t+1}, ...o_{t+K}$ following initial observation $o_t$ are drawn from the replay buffer in addition to the action distribution, reward and value prepared for MuZero loss. Then target latent state representation $s_{t+1,0}, ...s_{t+K,0}$ will be generated with $s_{t+k,0} = h(o_{t+k})$ for $k \in \{1, ...K\}$. At last unrolled state representation $s_{t,k}$ will be pulled toward target states representation $s_{t+k,0}$ by adding consistent loss on top MuZero loss. Then the EfficientZero loss corresponds to time step t is:

$$\mathcal{L}_{\text{EfficientZero}}(t) = \mathcal{L}_{\text{MuZero}}(t) + \sum_{k \in \{1, ...K\}} \mathcal{L}_{\text{SimSiam}}(s_{t,k}, s_{t+k,0})$$

By enforcing the consistent loss, EfficientZero assumes the unrolled latent state representation $s_{t,k}$ unrolled from initial latent $s_{t,0}$ with ground truth actions $a_t, ...a_{t+k-1}$ represents the ground truth state at time $t + k$.

We choose EfficientZero over SPR as the backbone of our method because SPR only uses the training signal of the prediction to stimulate the training of the representation network, while EfficientZero will be able to take advantage of higher quality unrolled states when performing the MCTS.

### 3.3 DATA AUGMENTATION

Augmentation has been an indispensable part of recent Imaged based Reinforcement Learning research. RAD (Laskin et al., 2020) has shown that data augmentation improves the sample efficiency and generalization of reinforcement learning. However, data augmentation could also harm the

performance of reinforcement learning. As shown in RAD, the performance of agents with augmentations like cutout are even worse than the baseline. This is because some critical information is removed from the image when doing augmentation. Due to this reason, the random shifts proposed by DrQ (Kostrikov et al., 2020; Yarats et al., 2021) have been the most popular data augmentation method since it makes the least change to the input observation while providing enough variance to regularize the representation network. Kostrikov et al. (2020) also pointed out that data augmentation can regularize downstream tasks like Q-learning beyond just regularizing the representation network.

We believe that even with the consistent loss, the augmentation is preserved at a certain level through the representation and dynamic network, affecting the prediction network. This is another reason why EfficientZero performs so well. So, when using augmentation, the notation of the latent state representations becomes:

$$\hat{s}_{t+k,0} = h(\hat{o}_{t+k})$$
$$\tilde{s}_{t+k,0} = h(\tilde{o}_{t+k})$$
$$\hat{s}_{t,k+1} = g_{\text{state}}(\hat{s}_{t,k}, a_{t+k})$$

Where ˆ and ˜ each represent an augmentation of a parameter at a time step $t$. $\hat{s}_{t,k}$ and $\tilde{s}_{t,k}$ should be similar but not necessarily the same.

### 3.4 SIMSIAM LOSS

SimSiam(Chen & He, 2020) is a state-of-the-art representation learning method. It maximize the similarity between two augmentations of one image. Compare to other representation learning methods like BYOL(Grill et al., 2020), SimCLR(Chen et al., 2020a), SwAV(Caron et al., 2020) and MoCo(He et al., 2019b; Chen et al., 2020b), SimSiam can perform very well even without negative sample pairs, large batch size and momentum encoder. SimSiam consist of a projector $P_1$ and a predictor $P_2$. Given a backbone representation function $h$ and two augmentations $\hat{x}$ and $\tilde{x}$ of same image $x$, the SimSiam loss is calcuated as follow.

$$\hat{y} = h(\hat{x})$$
$$\tilde{y} = h(\tilde{x})$$
$$\mathcal{D}(\hat{y}, \tilde{y}) = -\frac{\hat{y}}{||\hat{y}||} \cdot \frac{\tilde{y}}{||\tilde{y}||}$$

$$\mathcal{L}_{SimSiam}(\hat{y}, \tilde{y}) = \mathcal{D}(P_2(P_1(\hat{y})), stopgrad(P_1(\tilde{y}))) + \mathcal{D}(P_2(P_1(\tilde{y})), stopgrad(P_1(\hat{y})))$$

Where $\mathcal{D}$ is the negative cosine similarity. Since EfficientZero want to maximize the similarity between predicted latent state representation and ground truth future latent state representation instead of the similarity between two augmentation of one image, it don't need the symmetry. So the modified SimSiam loss of EfficientZero is as follow.

$$\mathcal{L}_{SimSiam}(\hat{s}_{t,k}, \tilde{s}_{t+k,0}) = \mathcal{D}(P_2(P_1(\hat{s}_{t,k})), stopgrad(P_1(\tilde{s}_{t+k,0})))$$

## 4 TEMPORAL CHANGE SENSITIVE REPRESENTATION

Most current representation algorithms focus on the general similarities and/or dissimilarities between different inputs. However, the difference between two observations is usually limited for image based reinforcement learning tasks. Especially when training a temporal predictive representation, the difference between two consecutive observations is only a small area in the image. Furthermore, if we consider the changes introduced by the augmentation, a few pixel differences could be easily ignored. Under this circumstance, enforcing the temporal consistency may not be enough for the changes to be preserved in the representation. So, we introduce temporal change sensitive representation (TCSR). In addition to enforcing the consistency of the representation, TCSR enforces the consistency of the change of the representation. The training pipeline is as shown in Figure 3.

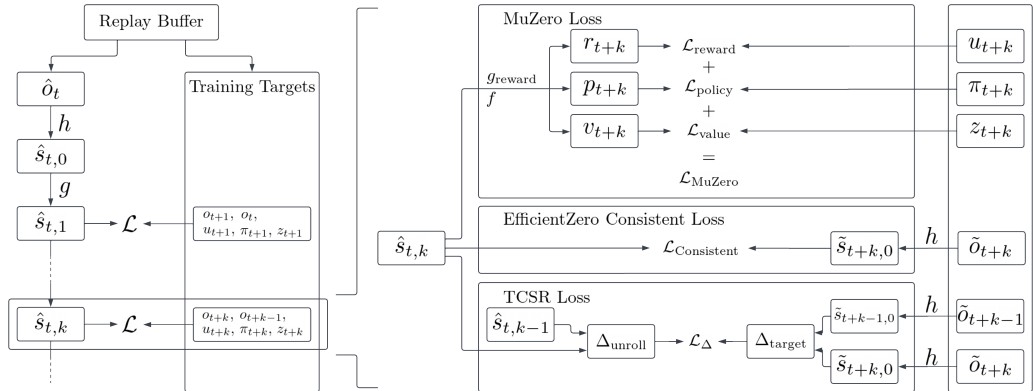

Figure 3: The training pipeline of Temporal Change Sensitive Representation(TCSR). On the left is the overall training pieline shared by MuZero, EfficientZero and TCSR. An inital observation $\hat{o}_t$ is drawn from the replay buffer and augmentated. Then $\hat{s}_{t,0}$ is obtained by encoding $\hat{o}_t$ with the representation network. Unrolled latent state representations $\hat{s}_{t,k}$ are generated iterately with the dynamic network. On the right shows the detile of how loss is calcuated for each unrolled latent state representation $\hat{s}_{t,k}$ in TCSR. $\hat{s}_{t,k-1}$ is the previous latent state representation that is used to generate $\hat{s}_{t,k}$ though not marked in the figure for conciseness. Both $\mathcal{L}_{\text{consistent}}$ and $\mathcal{L}_\Delta$ are SimSiam loss and they don't share parameters.

## 4.1 TCSR Loss

The TCSR Loss works alongside the training of EfficientZero as showen in 3. Consider the training pipeline of EfficientZero with augmentation. A time step $t$ is chosen, and corresponding information is retrieved from the replay buffer. Initial latent state representation is generated from augmented stacked observation with representation network $\hat{s}_{t,0} = h(\hat{o}_t)$, where $\hat{o}_t$ is augmented from the original observation $o_t$. Then unrolled latent state representations is generated iteratively as $\hat{s}_{t,k} = g_{\text{state}}(\hat{s}_{t,k-1}, a_{t+k-1})$ for $k \in \{1,..K\}$. The target latent state representations are generated from subsequence observations augmented with another parameter $\tilde{s}_{t+k} = h(\tilde{o}_{t+k})$ for $k \in \{1, \cdots K\}$, where $\tilde{o}_{t+k}$ is augmented from the original observation $o_{t+k}$. Note that $\hat{o}_t$ and $\tilde{o}_t$ are different augmentations of $o_t$ and $\hat{o}_{t+k}$ for $k \in \{0 \cdots K\}$ share same augmentation parameter (i.e. when the augmentation is random shift, they share the same shifting distance in x and y axis, etc.) We define the change of representation operation $\Delta$ as below.

$$\Delta_{\text{unroll}}(\hat{s}_{t,k}) = \hat{s}_{t,k} - \hat{s}_{t,k-1}$$
$$\Delta_{\text{target}}(\tilde{s}_{t+k,0}) = \tilde{s}_{t+k,0} - \tilde{s}_{t+k-1,0}$$

At last, the similarity loss between the difference of two consecutive unrolled latent state representations $\Delta_{\text{unroll}}(\hat{s}_{t,k})$ and the difference of two consecutive target latent state representation $\Delta_{\text{target}}(\tilde{s}_{t+k,0})$ is added on top of EfficientZero loss with a weight of $\lambda_{\text{TCSR}}$ to formulate TCSR loss.

$$\mathcal{L}_{\text{TCSR}}(t) = \mathcal{L}_{\text{EfficientZero}}(t) + \lambda_{\text{tcsr}} \sum_{k \in \{1, ...K\}} \mathcal{L}_{\text{SimSiam}}(\Delta_{\text{unroll}}(\hat{s}_{t,k}), \Delta_{\text{target}}(\tilde{s}_{t+k,0}))$$

Notice that the SimSiam loss here uses another set of projection and prediction net separated from the one used in EfficientZero loss.

The beauty of the TCSR loss is that the change of representation $\Delta$ is calculated from two representations generated by observations augmented with the same parameter. So the minor changes between two consecutive steps will not be overshadowed by the difference introduced by the augmentation of different parameters. At the same time, the augmentation is still able to regularize and generalize the training since $\Delta_{\text{unroll}}$ and $\Delta_{\text{target}}$ is generated from observation with different augmentation parameter.

Table 1: Scores of TCSR and other baselines on Atari100K benchmark. TCSR is 13.91% and 9.48% higher than the result of the EfficientZero source code that our method is base on. Note that the scores reported in the EfficientZero paper cannot be achieved by source code released on GitHub by the original author. So we choose to re-run the EfficientZero source code and report the result as our major baseline, which is still the SoTA algorithm on the Atari100K benchmark before our work.

| Game | Random | Human | SimPLe | CURL | DrQ | SPR | EfficientZero (paper reproted) | EfficientZero (Source code re-run) | TCSR |
|---|---|---|---|---|---|---|---|---|---|
| Alien | 227.8 | 7127.7 | 616.9 | 558.2 | 771.2 | **801.5** | 808.5 | 563.3 | 340.4 |
| Amidar | 5.8 | 1719.5 | 88 | 142.1 | 102.8 | **176.3** | 148.6 | 139.4 | 145.6 |
| Assault | 222.4 | 742 | 527.2 | 600.6 | 452.4 | 571 | 1263.1 | 1311.0 | **1946.7** |
| Asterix | 210 | 8503.3 | 1128.3 | 734.5 | 603.5 | 977.8 | 25557.8 | 9305.2 | **12389.3** |
| BankHeist | 14.2 | 753.1 | 34.2 | 131.6 | 168.9 | **380.9** | 351 | 242.9 | 264.7 |
| BattleZone | 2360 | 37187.5 | 5184.4 | 14870 | 12954 | **16651** | 13871.2 | 8432.3 | 13895.8 |
| Boxing | 0.1 | 12.1 | 9.1 | 1.2 | 6 | **35.8** | 52.7 | 21.4 | 21.8 |
| Breakout | 1.7 | 30.5 | 16.4 | 4.9 | 16.1 | 17.1 | 414.1 | 274.2 | **300.8** |
| ChopperCmd | 811 | 7387.8 | 1246.9 | 1058.5 | 780.3 | 974.8 | 1117.3 | **2312.5** | 1707.3 |
| CrazyClimber | 10780.5 | 35829.4 | 62583.6 | 12146.5 | 20516.5 | 42923.6 | 83940.2 | 72941.7 | **91619.3** |
| DemonAttack | 152.1 | 1971 | 208.1 | 817.6 | 1113.4 | 545.2 | 13003.9 | 5591.4 | **6043.5** |
| Freeway | 0 | 29.6 | 20.3 | **26.7** | 9.8 | 24.4 | 21.8 | 3.6 | 3.6 |
| Frostbite | 65.2 | 4334.7 | 254.7 | 1181.3 | 331.1 | **1821.5** | 296.3 | 258.0 | 253.4 |
| Gopher | 257.6 | 2412.5 | 771 | 669.3 | 636.3 | 715.2 | 3260.3 | 1568.2 | **1712.1** |
| Hero | 1027 | 30826.4 | 2656.6 | 6279.3 | 3736.3 | 7019.2 | 9315.9 | 8689.8 | **10817.0** |
| Jamesbond | 29 | 302.8 | 125.3 | **471** | 236 | 365.4 | 517 | 255.2 | 441.7 |
| Kangaroo | 52 | 3035 | 323.1 | 872.5 | 940.6 | **3276.4** | 724.1 | 1415.6 | 1541.1 |
| Krull | 1598 | 2665.5 | 4539.9 | 4229.6 | 4018.1 | 3688.9 | 5663.3 | 6505.9 | **6684.6** |
| KungFuMaster | 258.5 | 22736.3 | 17257.2 | 14307.8 | 9111 | 13192.7 | 30944.8 | 20949.0 | **22663.9** |
| MsPacman | 307.3 | 6951.6 | **1480** | 1465.5 | 960.5 | 1313.2 | 1281.2 | 1151.6 | 1033.6 |
| Pong | -20.7 | 14.6 | 12.8 | -16.5 | -8.5 | -5.9 | 20.1 | 12.7 | **17.7** |
| PrivateEye | 24.9 | 69571.3 | 58.3 | -13.6 | 124 | 96.7 | 100.0 | 17.6 | |
| Qbert | 163.9 | 13455 | 1288.8 | 1042.4 | 854.4 | 669.1 | 13781.9 | 6006.4 | **7192.1** |
| RoadRunner | 11.5 | 7845 | 5640.6 | 5661 | 8895.1 | 14220.5 | 17751.3 | 14213.5 | **15862.5** |
| Seaquest | 68.4 | 42054.7 | 683.3 | 384.5 | 301.2 | 583.1 | 1100.2 | **1017.4** | 690.4 |
| UpNDown | 533.4 | 11693.2 | 3350.3 | 2955.2 | 3180.8 | **28138.5** | 17264.2 | 5010.5 | 8081.5 |
| Normed Mean | 0 | 1 | 0.443 | 0.381 | 0.357 | 0.704 | 1.943 | 1.243 | **1.451** |
| Normed Median | 0 | 1 | 0.144 | 0.175 | 0.268 | 0.415 | 1.09 | 0.448 | **0.602** |

## 5 EXPERIMENT

### 5.1 ENVIRONMENT AND BASELINE

We evaluate TCSR on Atari100k (Kaiser et al., 2019), a widely used benchmark for sample efficient reinforcement learning. The reinforcement learning agent is allowed to interact and collect 100,000 steps with a frame skipping of 4. So a total of 400,000 frames are generated from the simulator. Atari100K is mostly used to test the sample efficiency of reinforcement learning algorithms (Schwarzer et al., 2020; Kostrikov et al., 2020; Ye et al., 2021). We follow the same settings of EfficientZero (Ye et al., 2021) to perform the evaluation. For each task, we perform 6 runs with different seeds and each run with 32 evaluation episodes. The mean accumulated rewards of 6 runs are calculated and recorded as the raw score. Then the human normalized score is calculated for each task with the following equation.

$$\text{score}_{\text{normed}} = \frac{\text{score}_{\text{raw}} - \text{score}_{\text{random}}}{\text{score}_{\text{human}} - \text{score}_{\text{random}}}$$

At last, the mean and median normed score of 26 atari games are used to evaluate the overall performance of this reinforcement learning agent. Our method is built on the latest source code released by EfficientZero author on GitHub. We could not reproduce the result reported in the EfficientZero paper using the code released by the author. Though the result from running the author released code is still better than other baselines such as SPR (Schwarzer et al., 2020). Since our change is on top of the EfficientZero source code, we believe it is fair to compare the result of our method with the result from running the EfficientZero source code. We have run the latest source code released by the EfficientZero author on our machine and report the result in Table 1. The re-run result achieves SoTA performance on 12 out of 26 games in the Atari100K benchmark and achieves mean and median human normalized score of 1.243 and 0.448, which is still the SoTA method on the Atari100K benchmark before our work. We have also included other popular algorithms(Schwarzer

et al., 2020; Kostrikov et al., 2020; Kaiser et al., 2019; Srinivas et al., 2020) as baseline to compare with TCSR in Table 1.

## 5.2 RESULTS

The result of TCSR on the Atari100K benchmark is shown in Table 1. Our method achieves the highest score in 12 out of 26 games. Human normalized score wise, TCSR achieves a high score of 1.451 mean and 0.57 median, which are 16.73% and 34.24% higher than the result of the EfficientZero source code that our method is based on. Specificaly, TCSR outperformance EfficientZero in 19 out of 26 games in Atari100K benchmark as shown in 2.

## 5.3 VISUALIZATION

To understand how TCSR influences the learned latent state representation, we trained a decoder network to visualize what information is contained in the latent state. We only train the decoder with latent state representation generated directly from the representation network and use ground truth observation as the target. This ensures that the decoder can only reconstruct information contained in the current latent state representation and cannot predict. Mean square error is used to calculate the loss at pixel lever. We stop the gradient at the latent state representation so that the training of image reconstruction will not affect the regular training pipeline. When visualizing, we feed EfficientZero and TCSR with the same observation and action sequence to see the difference in unrolled latent states. An example of the Atari game Assault reconstruction result is shown in Figure 1. In Assault, a UFO will release enemies to attack the fighter controlled by the agent. When no enemy is on the screen, the UFO will release another enemy immediately. However, when enemies are on the screen, the UFO will release a new enemy under a certain rule. TCSR can capture such information and correctly predict when and where the UFO will release the new enemy. This explains why TCSR outperforms EfficientZero in 17 out of 26 tasks in the Atari100k benchmark.

## 6 CONCLUSION

This paper presented Temporal Change Sensitive Representation (TCSR), a self-supervised auxiliary task designed for reinforcement learning algorithms that train a dynamic model. We enforce the temporal difference between unrolled latent state representations to be consistent with the temporal difference between target latent state representations. Calculating the difference between two consecutive states is similar to taking derivative, which is a common practice in the field of Mathematics and Physics when trying to learn the dynamics. The results show that our method can help agents better capture critical information in the latent state representations and better unroll those latent state representations. With the help of representation learned by TCSR, the EfficientZero backbone is able to achieve state-of-the-art performance in the Atari100K benchmark. Possible future extension of TCSR includes applying it to different losses other than SimSiam and higher order of temporal differences.

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

Table 2: Hyper-parameters for TCSR

| Parameter | Setting |
|---|---|
| Observation down-sampling | $96 \times 96$ |
| Frames stacked | 4 |
| Frames skip | 4 |
| Reward clipping | True |
| Terminal on loss of life | True |
| Max frames per training episode | 12k |
| Max frames per evaluate episode | 108K |
| Discount factor | $0.997^4$ |
| Minibatch size | 256 |
| Optimizer | SGD |
| Optimizer: learning rate | 0.2 |
| Optimizer: momentum | 0.9 |
| Optimizer: weight decay (c) | 0.0001 |
| Max gradient norm | 5 |
| Priority exponent ($\alpha$) | 0.6 |
| Priority correction ($\beta$) | $0.4 \to 1$ |
| Training steps | 120K |
| Evaluation episodes | 32 |
| Min replay size for sampling | 2000 |
| Self-play network updating inerval | 100 |
| Target network updating interval | 200 |
| Unroll steps | 5 |
| TD steps | 5 |
| Policy loss coefficient | 1 |
| Value loss coefficient | 0.25 |
| Self-supervised consistency loss coefficient | 2 |
| Temporal change consistency loss coefficient | 0.2 |
| Recurrent gradient scaling factor | 1.0 |
| LSTM horizontal length | 5 |
| Dirichlet noise ratio | 0.3 |
| Number of simulations in MCTS | 50 |
| Reanalyzed policy ratio | 0.99 |

## A    IMPLEMENTATION DETAIL

TCSR is implemented on the EfficientZero source code. We followed EfficientZero's implementation on the representation network, dynamic network, reward prediction network, value prediction network and policy prediction network. TCSR has one addition set of SimSiam predictor and projector that have the exact same architecture as those in EfficientZero. They are used to maximize the similarity between the difference of two consecutive unrolled latent state representation and the difference of two consecutive target latent state representation.

The architecture of predictor in SimSiam network is as follow:

- 1 fully connected layers and 256 output dimensions. (BN + ReLU)
- 1 fully connected layers and 256 output dimensions. (BN + ReLU)
- 1 fully connected layers and 256 output dimensions. (BN)

The architecture of projector of SimSiam network is as follow:

- 1 fully connected layers and 64 output dimensions. (BN + ReLU)
- 1 fully connected layers and 256 output dimensions.

Some update had been made to the Hyper-parameters in the EfficientZero source code since the publication of EfficientZero paper. TCSR also made minor adjustment to the Hyper-parameters. The Hyper-parameters are as shown in table 2. The learning rate schedule is as showen below.

$$\text{learning rate} = \begin{cases} 0.0002 \cdot \text{step}, & 0 < \text{step} \leq 1000 \\ 0.2, & 1000 < \text{step} \leq 100K \\ 0.02 & 100K < \text{step} \leq 120K \end{cases}$$

## B    DATA AUGMENTATION

We followed EfficientZero on the implementation of data augmentation. Some details regarding the implementation of data augmentation isn't mentioned in EfficientZero paper, but we believe these

details are extra important for TCSR. So we will discuss them here. The data augmentation of EfficientZero consist two part, random shift and intensity, which is defined as follow.

**Random Shift**
Pad the input image of size 96 × 96 to 108 x 108 by repeating boundary pixels for 4 pixels. Then random crop the image back to 96 x 96.

**Intensity**
Generate a random scalar $s$ by $s = 1 + 0.05 \cdot clamp(r, -2, 2)$, where $r \sim \mathcal{N}(0, 1)$. Multiply the value on each pixel of input image by $s$

In the main part of this paper, we mentioned a concept called the "parameter" of augmentation. By "parameter" we mean the crop position and the value of scalar $s$. We say $\hat{o}_t$ and $\hat{o}_{t+1}$ share the same augmentation parameter when they share the same crop position and scalar $s$. We say $\hat{o}_t$ and $\tilde{o}_t$ have different augmentation parameter when the crop position and scalar are sampled separately.

The "parameter" of augmentation is extra important for TCSR. Suppose we have two consecutive observation $o_t$ and $o_{t+1}$. We augment them with different augmentation to get $\hat{o}_t$ and $\tilde{o}_{t+1}$ and encode them with the representation network $h$ to get $\hat{s}_t$ and $\tilde{s}_{t+1,0}$. If we calculate the difference between $\hat{s}_t$ and $\tilde{s}_{t+1,0}$, the result is meaningless since the difference introduced by the augmentation could be bigger than the temporal change.

## C  DECODER

A decoder is used to demonstrate the quality of unrolled latent state representation predicted by TCSR. The architecture of the decoder is as follow:

- 1 transposed residual block with 64 planes. (BN + ReLU)
- Up sample with nn.functional.interpolate, output size 12 x 12, mode: nearest.
- 1 transposed residual block with 64 planes. (BN + ReLU)
- Up sample with nn.functional.interpolate, output size 12 x 12, mode: nearest.
- 1 transposed residual block with 64 planes. (BN + ReLU)
- 1 transposed upsample residual block with stride 2 and 32 output planes (BN + ReLU)
- 1 transposed residual block with 32 planes. (BN + ReLU)
- 1 transposed upsample residual block with stride 2 and 12 output planes (BN + ReLU)

