# OpenReview forum: "Temporal Change Sensitive Representation for Reinforcement Learing"
_ICLR.cc/2023/Conference — Submitted to ICLR 2023_

### Official Review · Reviewer_1Xom · 2022-10-21

**Confidence:** 3
**Correctness:** 1
**Technical Novelty And Significance:** 3
**Empirical Novelty And Significance:** 2
**Recommendation:** 3

**Clarity, Quality, Novelty And Reproducibility:**

The quality is poor, primarily due to the flaw with the main contribution discussed above.  The novelty seems to be slightly limited, but is adequate.  Regarding clarity: the paper is well-written in general, but has some severe clarity issues in a few places, including undefined terms (see above).

**Strength And Weaknesses:**

This is an interesting and well-motivated paper studying an important topic.

However, the contribution is entirely empirical (the contribution boils down to the proposal of TCSR, and the claim that TCSR “achieves SoTA performance”), and the experiments are deeply flawed.  The results are based on 3 runs, which is unacceptable in RL, due to the high variance between runs.  This topic has been repeatedly covered in publications, talks, etc. in recent years, so I will not elaborate further.  It is interesting to notice, however, the fact that the “reported EfficientZero” score greatly outperforms the TCSR score.  The large discrepancy between the “reported EfficientZero” score and the “source code re-run EfficientZero” score likely comes from the low number of runs used.  This discrepancy is an excellent illustration of the problem with only using 3 runs.

Clarity issues:
- The loss functions that are in the definitions of the MuZero loss are not defined (nor is it easy to infer what they are).
- Similarly, the SimSiam loss is not defined.
- g_{state} is not defined.
- The difference between s^ and s~ (defined at the end of section 3) is not clear.  They “each represent an augmentation of a parameter”.  So there are two parameters to augment?  (I do not think that’s correct, but I do not know how else to read it.)

Minor edits:
- “reward is sparse(Shelhamer et al., 2016)” (missing a space)
- “The training pipeline is as shown in Figure 3” (missing period)

**Summary Of The Paper:**

The authors propose TCSR; they perform experiments to show that it improves upon EfficientZero and achieves SOTA performance on the Atari100K benchmark.


**Summary Of The Review:**

This is an interesting and well-motivated paper studying an important topic.  However, the contribution is entirely empirical, and the experiments are deeply flawed.

---

> ### Author Response · Authors · 2022-11-19
> **Response to Reviewer 1Xom**
>
> Thank you for the constructive feedback and insightful comments. The clarity issues have been dressed in the general response. Here are the responses regarding the problems of the experiment.
>
> Reviewer:
>
> The results are based on 3 runs, which is unacceptable in RL, due to the high variance between runs. This topic has been repeatedly covered in publications, talks, etc. in recent years, so I will not elaborate further. It is interesting to notice, however, the fact that the “reported EfficientZero” score greatly outperforms the TCSR score. The large discrepancy between the “reported EfficientZero” score and the “source code re-run EfficientZero” score likely comes from the low number of runs used. This discrepancy is an excellent illustration of the problem with only using 3 runs.
>
> Authors:
>
> We totally agree with the reviewer that the result based on 3 runs has a high variance. So we have done more experiments to increase the number of random seeds from 3 to 6. There are some minor changes to the scores of both EfficientZero and TCSR. However, the discrepancy between the "reported EfficientZero" score and the "source code re-run EfficientZero" remain unchanged.

---

### Official Review · Reviewer_JdZQ · 2022-10-24

**Confidence:** 4
**Correctness:** 3
**Technical Novelty And Significance:** 2
**Empirical Novelty And Significance:** 1
**Recommendation:** 3

**Clarity, Quality, Novelty And Reproducibility:**

Clarity:
- Although the content of the paper is relatively small, it is difficult to understand the proposed method without knowing the related work in detail, since the paper not at all self-contained. For example, not even the actual loss (the main contribution) is shown in the paper; instead, the reader is assumed to know $\mathcal{L}_{\text{SimSiam}}$. Furthermore, the difference between the policy $\pi$ and $p$ is not clear as they have not been properly defined. The paper should discuss EfficientZero and the SimSiam loss in more detail, which could have been done in the current submission, given that it only used a bit more than seven pages.
- The paper is not well-structured. For example, Fig. 1 is mentioned in the Appendix, but shows a comparison with EfficientZero which has not been mentioned at that point. The actual Figure is only described in the experiments section at the end of the paper.
- Also the writing is quite sloppy and even contains errors that could have been automatically found with a spell check ("represnetation").
Several sentences even have multiple mistakes.

Quality and Reproducibility:
- The paper contains little information on how the experimental study has been conducted. The paper does not come with any appendix nor source code and is therefore not at all reproducible. It is not reported which hyperparameters and network architectures have been used and how they have been tuned.
- For many environments, the proposed method performed worse than the previously reported results of EfficientZero. While the paper argues that the comparison is fair by using the same implementation (of the original authors of EfficientZero) for evaluating the new method and for rerunning EfficientZero, it seems like the authors of the current submission did not even try to reproduce the reported EfficientZero results beyond running the code on their own machine.

Novelty:
- The novelty of the paper lies given only by a new loss term, which was not well motivated.


**Strength And Weaknesses:**

Strength:
- The paper presents a novel and reasonable auxiliary, that achieves good empirical results.

Weaknesses:
- The contribution is rather small, as the modification of the previous method is only incremental and not theoretically justified.
- The presentation is sloppy.

**Summary Of The Paper:**

The paper proposes a new auxiliary loss for learning representations for reinforcement learning from images. The loss penalizes the difference between the change of two subsequent latent states for one image augmentation and the change of two subsequent states for a different augmentation. By integrating this loss into "EfficientZero" (Ye et al. 2021) the performance on Atari benchmarks good be improved.


**Summary Of The Review:**

The contribution of the paper is incremental, but could be somewhat interesting.
However, the communication of the work is substandard, and the paper would require a major revision by:
- describing the proposed method in a more self-contained way,
- providing an in-depth description on the experimental procedure,
- improving the structure of the paper,
- and, ideally, investigating the discrepancy between the reported results of EfficientZero, and the results obtained in the current experiment (I recommend contacting the authors).
Furthermore, providing the source code for reproducing the experiments would be highly appreciated.

---

> ### Author Response · Authors · 2022-11-19
> **Response to Reviewer JdZQ**
>
> Thank you for the constructive feedback and insightful comments. For problems regarding quality and reproducibility, please check out the general response. Here are the responses regarding other problems you have pointed out.
>
> Reviewer:
>
> Although the content of the paper is relatively small, it is difficult to understand the proposed method without knowing the related work in detail, since the paper not at all self-contained. For example, not even the actual loss (the main contribution) is shown in the paper; instead, the reader is assumed to know $\mathcal{L}_\text{SimSiam}$. Furthermore, the difference between the policy $\pi$ and $p$ is not clear as they have not been properly defined. The paper should discuss EfficientZero and the SimSiam loss in more detail, which could have been done in the current submission, given that it only used a bit more than seven pages.
>
> Authors:
>
> Thanks for pointing out that the paper miss explanation on $\mathcal{L}_\text{SimSiam}$ and RL losses in MuZero. We have added section 3.4 to discuss $\mathcal{L}_\text{SimSiam}$. We also added information about reward loss, policy loss and value loss in section 3.1. $\pi$ is the distribution of action at the root of MCTS and $p$ is the predicted policy, which is mentioned in section 3.1. We have also updated Figure 3, demonstrating how MuZero and EfficientZero train the representation and dynamic networks. There is space for further explanation on MuZero or EfficientZero, but it would make the paper unbalanced since this paper focuses on improving the predicted latent state representation quality.

---

### Official Review · Reviewer_yZGc · 2022-10-25

**Confidence:** 3
**Correctness:** 3
**Technical Novelty And Significance:** 2
**Empirical Novelty And Significance:** 2
**Recommendation:** 3

**Clarity, Quality, Novelty And Reproducibility:**

The paper is written clearly, and the results/experiments contain adequate details to be convincing.

**Strength And Weaknesses:**

Strength:

The paper is well-organized. Each section does what it aims to do. The algorithms are described very clearly.

Weaknesses:

1. The extent of novelty is relatively low. On a high level, the paper appears to be a combination of the existing EfficientZero algorithm(a temporal-aware variant of the MuZero algorithm over non supervised contents, with data augmentation and minor adjustments over similarity losses. More elaboration on novelty would be helpful.

2. The motivation should be stated more clearly. While the experiments are adequate, there should be more explanation as to how/why each set of experiments correspond to the context/setting of the problem. Additionally, it would be helpful to provide some visual illustrations(or other qualitative demonstrations) as to why the algorithm works, as opposed to simply listing the performances SOTA and conclude accordingly.

3. Would it be possible to incorporate experiments on other datasets? Empirical results solely on Atari-based datasets seem a bit inadequate, especially when the algorithm seeks to address representational learning for reinforcement learning problems in general.

**Summary Of The Paper:**

This paper proposes Temporal Change Sensitive Representation, a self-supervised representation algorithm that seeks to capture the temporal representation changes in RL. Experiments on different datasets suggest that the proposed algorithm does outperform benchmark SOTAs in the Atari100k environment.

**Summary Of The Review:**

My rating would be a 3, though I'm happy to adjust my score if the authors address my concern.

---

> ### Author Response · Authors · 2022-11-19
> **Response to Reviewer yZGc**
>
> We thank you for the constructive feedback and insightful comments that help improved the quality of the paper from several aspects. Here are the responses regarding the problems you have pointed out.
>
> Reviewer:
>
> The extent of novelty is relatively low. On a high level, the paper appears to be a combination of the existing EfficientZero algorithm(a temporal-aware variant of the MuZero algorithm over non supervised contents, with data augmentation and minor adjustments over similarity losses. More elaboration on novelty would be helpful.
>
> Authors:
>
> We agree that the change from EfficientZero to TCSR is simple, but the novelty is not small. Our method forces the dynamic network to focus on the small changes in the latent state representation between two consecutive timesteps, which would be ignored otherwise if only use similarity based loss. One example is Figure 1 in the paper. EfficientZero fails to predict the enemy's position, while TCSR accurately predicts the enemy's position. This is critical since MuZero/EfficientZero backbone relies on the predicted latent state representation to choose action. On the other hand, simplicity means our method is generalizable. Our method can be added on top of any method that learns a dynamic network.
>
> Reviewer:
>
> The motivation should be stated more clearly. While the experiments are adequate, there should be more explanation as to how/why each set of experiments correspond to the context/setting of the problem. Additionally, it would be helpful to provide some visual illustrations(or other qualitative demonstrations) as to why the algorithm works, as opposed to simply listing the performances SOTA and conclude accordingly.
>
> Authors:
>
> Thanks for the suggestion. Our motivation is very simple, which is to improve the quality of the predicted latent state representation. A high quality predicted latent state representation should contain information that is related to the task. To illustrate if our predicted latent state representation contains information related to the task, we trained a decoder to reconstruct the future observation from the predicted latent state representation, as shown in Figure 1 and discussed in section 5.3.

---

### Official Review · Reviewer_VxAy · 2022-10-25

**Confidence:** 3
**Correctness:** 2
**Technical Novelty And Significance:** 2
**Empirical Novelty And Significance:** 2
**Recommendation:** 3

**Clarity, Quality, Novelty And Reproducibility:**

The paper is not clear at many places and the proposed method is not detailed enough. No implementation details are provided which makes the paper not reproducible.

**Strength And Weaknesses:**

Weaknesses:
The paper has several weakness in writing, results, explanation of the approach. See below:

Results:
- The paper does not evaluate the proposed method with significant experiments. Only Table 1 shows results on Atart100K which shows improvement in 11 out of 26 games as compared to the author's claim of 17 out of 26. These results are also not consistent with Figure 2. The table results are also not explained.
- The visualization shown Figure 1 is highly basic and any encoder should be able to encode the representations. Why many method are failing to encode it is not explained anywhere in the paper.
- Training details are not provided, the results seems not reproducible with any implementation details.
- The author's claim self-supervision. I didn't find any information which help TCSR to train in a self-supervised manner.

Method:
- The approach is not detailed. The reader cannot understand the proposed method as many details are missing or not explained. For example, Figure 3 is never explained and it is the training pipeline. I am not able to infer what is the training part in this figure.
- The notations are not clear. In s_t^k, s is state at time t but what is k?
- The authors gave a good emphasis on consistent loss but it is not explained. an equation mentioned it in sec 3.2 but what is the loss? The equations are not numbered.
- The TCSR loss in sec 4.1 is not clear. Why t+k time is the target for state at time t?

Writing:
- Sec 3.1, first paragraph third line, encodes a observation
- Sec 3.2, represnetation
- Full stop in last line of page 5, many places in sec 4.1.
- Table:1 can be written as Table 1.

**Summary Of The Paper:**

The paper presents a method named TCSR which is based on EfficientZero. TCSR helps a reinforcement learning agent to learn better latent representations in a self supervised manner.

**Summary Of The Review:**

The paper has several weaknesses and it is not ready for publication. See weaknesses for justification.

---

> ### Author Response · Authors · 2022-11-19
> **Response to Reviewer VxAy**
>
> Thank you for the constructive feedback and insightful comments that help improved the quality of the paper from several aspects. Here are the responses regarding the problems you have pointed out.
>
> Reviewer:
>
> The paper does not evaluate the proposed method with significant experiments. Only Table 1 shows results on Atart100K which shows improvement in 11 out of 26 games as compared to the author's claim of 17 out of 26. These results are also not consistent with Figure 2. The table results are also not explained.
>
> Authors:
>
> The numbers have been updated due to the increase in the number of runs. Though we never claimed our method achieved the best score in 17 out of 26 games in the Atari100K benchmark. The claim was TCSR outperformance EfficientZero in 17 out of 26 games in the Atari100K benchmark.
>
> Reviewer:
>
> The visualization shown Figure 1 is highly basic and any encoder should be able to encode the representations. Why many method are failing to encode it is not explained anywhere in the paper.
>
> Authors:
>
> It is indeed that in a standard encoder and decoder setup, reconstructing the observation is basic. However, the backbone we use doesn't rely on reconstruction loss. So to prevent the training signal of the decoder from influencing the overall training, we detached the latent state representation before sending it to the decoder. In addition to that, to visualize the improvement of the dynamic network with TCSR loss, we have never used predicted latent state representation to train the decoder. The decoder is only trained with latent state representation. This means if the observation reconstructed from the predicted latent state is consistent with ground truth observation, the prediction made by the dynamic network is accurate, and vice versa. There is similar visualization(Figure 4) in the EfficientZero paper. The quality of reconstructed observation on the long horizon is very low. So reconstructing a high-quality observation in this setup is far from trivial.
>
> In case you and future readers are interested, we have also added the architecture of our decoder and detailed training method in the Appendix. Most of the discussion above can also be found in section 5.3 of our paper.
>
> Reviewer:
>
> The notations are not clear. In s_t^k, s is state at time t but what is k?
> The TCSR loss in sec 4.1 is not clear. Why t+k time is the target for state at time t?
>
> Authors:
>
> MuZero, EfficientZero and TCSR all train the representation network and the dynamic network at the same time. To train the dynamic network, we need data from future steps. $k$ means the current online latent state representation has been unrolled by the dynamic network for $k$ times and therefore needs data from $t+k$ step.

---

### Author Response · Authors · 2022-11-19
**General Response**

We thank the reviewers for their efforts in reviewing the paper and their insightful feedback that help to improve the paper. We are pleased that all reviewers showed interest in our work. The following adjustments are made to address some common issues proposed by reviewers.

- Increase the number of runs from 3 to 6 for both "EfficientZero source code" and TCSR for a more accurate result. Corresponding data are updated throughout the paper.
- Redraw Figure 3 so that reviewers and future readers can clearly understand how TCSR loss works and its relation with EfficientZero and MuZero.  This Figure also demonstrates how EfficientZero and MuZero are trained.
- Added the detail of SimSiam Loss in section 3.4, which is used in both EfficientZero and TCSR.
- Added Implementation Detail in Appendix A for Reproducibility.
- Added detail of data augmentation in Appendix B to address the question: What is the parameter of an augmentation?
- Added the architecture of the decoder that is used for visualization.


We also found that most reviewers showed concern regarding the comparison between the score of TCSR and the score of "EfficientZero source code re-run" rather than the score of "EfficientZero paper reported". We understand that this is not a common practice. For us, it is a harsh decision. Re-running the whole benchmark is expensive and time consuming. But we have tried to communicate with the author regarding the reproducibility of EfficientZero, and we never received any response. Another paper on arxiv(https://arxiv.org/abs/2210.10763) reported scores of "EfficientZero source code re-run" of selected games in Atari100K Benchmark, which is also inconsistent with the score reported in the original paper. There are also many issues (#21 #23 #32 in https://github.com/YeWR/EfficientZero/issues) in the EfficientZero GitHub repository regarding the reproducibility of EfficientZero. None received any response from the author.

Based on the above reasons, re-running the source code provided by the EfficientZero author makes a fair comparison. Despite the above problems, the "EfficientZero source code re-run" score still has state-of-the-art performance in the Atari100K benchmark before our work. It achieves the highest score in 12 out of 26 games in Atari100K benchmark among the 5 baselines selected in the paper and achieves mean and median human normalized scores of 1.243 and 0.448. As a reference, SPR achieves the highest score in 5 out of 26 games and mean and median human normalized scores of 0.704 and 0.415. So, EfficientZero is a solid foundation for future research.

---

> ### Author Response · Authors · 2022-11-21
> **A Summary of Score Change after Increasing the Number of Random Seed from 3 to 6**
>
> To accommodate the high variance of scores between runs of RL algorithms and increase the accuracy of our result, we have performed additional experiments and increased the number of random seeds from 3 to 6 for both EfficientZero and TCSR. In general, the conclusion remains unchanged. TCSR is the new SoTA method in the Atari100K benchmark and outperforms EfficientZero. Specifically, in the Atari100K benchmark, TCSR's scores are highest in 12 out of 26 games among selected baselines (11 out of 16 games when using 3 seeds). Compared with EfficientZero, TCSR achieves a higher score in 19 out of 26 games (17 out of 26 games when using 3 seeds). The changes in human normalized scores are as follows:
>
> | Game            | EfficientZero | EfficientZero | TCSR  | TCSR  |
> |-----------------|---------------|---------------|-------|-------|
> | Number of Seeds | 3             | 6             | 3     | 6     |
> | Normed Mean     | 1.221         | 1.243         | 1.414 | 1.451 |
> | Normed Median   | 0.520         | 0.448         | 0.570 | 0.602 |
>
> At last, it is important to note that though the TCSR loss proposed in our paper is simple, it increases the performance of EfficientZero by a significant margin. Especially when you consider that the increase in performance from EfficientZero to TCSR is comparable to the increase in performance from SPR to EfficientZero, where EfficientZero is much more complex than SPR. A simple but powerful method should be welcomed because it is easier to be applied in any other backbone, therefore potentially benefiting more future research and application.

---

### Decision · Program_Chairs · 2023-01-20

**Decision:**

Reject

**Justification For Why Not Higher Score:**

Writing is poor. Experimental analysis is narrow in scope. Novelty is limited.

**Justification For Why Not Lower Score:**

N/A

**Metareview: Summary, Strengths And Weaknesses:**

The paper presents a new self-supervised RL algorithms called TCSR. It builds on top of efficientNet for the Atari100k benchmark suite. The experimental results in this paper on Atari100k are quite strong. However, there are several concerns from the reviewers. The novelty of this work is limited as it is a combination of EfficientZero and data augmentation-based similarity. Hence we would need to see strong experimental results. However, on the experimental front, it is quite unclear on what is the context of the experiments. The writing and discussion needs to be significantly improved to conveys the key insights and methodology. Finally, Atari100k is quite a narrow set of tasks. We recommend experimenting with a broader set of tasks, perhaps including the DM Control tasks.